# The Current Status of DNA-Repair-Directed Precision Oncology Strategies in Epithelial Ovarian Cancers

**DOI:** 10.3390/ijms24087293

**Published:** 2023-04-14

**Authors:** Hiu Tang, Sanat Kulkarni, Christina Peters, Jasper Eddison, Maryam Al-Ani, Srinivasan Madhusudan

**Affiliations:** 1Department of Oncology, Nottingham University Hospitals, Nottingham NG5 1PB, UK; 2Department of Medicine, Sandwell and West Birmingham Hospitals, Lyndon, West Bromwich B71 4HJ, UK; 3Department of Oncology, Sussex Cancer Centre, University Hospitals Sussex NHS Foundation Trust, Brighton BN2 5BD, UK; 4College of Medical & Dental Sciences, University of Birmingham Medical School, Birmingham B15 2TT, UK; 5Nottingham Biodiscovery Institute, School of Medicine, University of Nottingham, University Park, Nottingham NG7 3RD, UK

**Keywords:** DNA repair, ovarian cancer, synthetic lethality, precision oncology, PARPi, BRCA

## Abstract

Survival outcomes for patients with advanced ovarian cancer remain poor despite advances in chemotherapy and surgery. Platinum-based systemic chemotherapy can result in a response rate of up to 80%, but most patients will have recurrence and die from the disease. Recently, the DNA-repair-directed precision oncology strategy has generated hope for patients. The clinical use of poly(ADP-ribose) polymerase (PARP) inhibitors in BRCA germ-line-deficient and/or platinum-sensitive epithelial ovarian cancers has improved survival. However, the emergence of resistance is an ongoing clinical challenge. Here, we review the current clinical state of PARP inhibitors and other clinically viable targeted approaches in epithelial ovarian cancers.

## 1. Introduction

Ovarian cancer is the fifth most common cancer in women, with over 7500 women diagnosed each year in the United Kingdom [1]. Epithelial ovarian cancer (EOC) is the most common histological type of ovarian cancer, with around 75% of cases diagnosed at FIGO stage III–IV due to symptoms being vague, ill-defined, and often attributed to benign conditions. The risk factors for ovarian cancer include increasing age, positive family history, increasing age of reproduction, high socioeconomic classes, nulliparity, and obesity [2]. Surgery continues to have a central role in the treatment of EOC in combination with chemotherapy. For advanced-stage epithelial ovarian cancer, the standard treatment with optimal surgery and chemotherapy generates a median progression-free survival of 22.4 months as per the ICON-7 clinical trial [3]. The outlook of EOC has improved since the introduction of precision oncology strategies targeting tumour angiogenesis and DNA repair, as discussed later in the review.

### 1.1. The Biology of DNA Repair Pathways

DNA is constantly under attack from endogenous sources, such as reactive oxygen species (ROS) or replication errors, and exogenous sources, such as ionising radiation (IR) and chemotherapy agents. Consequently, to maintain the integrity of the genome, both prokaryotes and eukaryotes have evolved highly conserved DNA damage response (DDR) pathways to identify and correct DNA damage [4]. However, not all DNA damage is necessarily repaired. Dependent on the type of damage, cells may utilise different pathways, which can result in: tolerance to the damage, transcriptional activation, cell cycle arrest, apoptosis, or the repair of the lesion [5,6].

#### 1.1.1. DNA Repair Pathways

Human cells have evolved at least six major repair pathways dependent on the type of damage sustained (Figure 1), although there is a crossover in effector proteins between pathways. The key targeted repair pathways are outlined below but are more comprehensively reviewed elsewhere [6].

#### 1.1.2. Direct Reversal

In humans, some alkylating DNA lesions, which can occur following treatment with alkylating agents (a common systemic anticancer therapy) [7], can be directly reversed in situ by the sacrificial enzymes O^6^-alkylguanine-DNA alkyltransferase (AGT) or methylguanine methyltransferase (MGMT) [8]. Consequently, normal or higher tumoural MGMT levels are negatively associated with patient outcomes due to greater alkylating agent resistance [9]. Alternatively, alkylating lesions can be repaired through oxidative reversal with the AlkB dioxygenases (ABH2 and ABH3) [8,10], or in some cases, through base excision repair (BER) [11].

#### 1.1.3. Base Excision Repair (BER)

The BER pathway is primarily responsible for repairing smaller, nondistorting single-strand damage or breaks [12,13], typically as a result of ROS, spontaneous deamination, and IR [14]. BER consists of two distinct pathways: DNA polymerase beta (polβ)-mediated short-patch repair of single nucleotides or proliferating cell nuclear antigen (PCNA)-dependent long-patch repair of 2–6 nucleotides [11,15]. Both pathways begin with the removal of the damaged base by damage-specific DNA glycosylases generating an abasic site (AP-site) and the incision of the DNA backbone by AP-endonuclease 1 (APE1). The subsequent excision of remaining fragments and the insertion of the correct base are performed by polβ in conjunction with X-ray cross-complementing group 1 protein (XRCC1) in short-patch repair or flap-endonuclease 1 (FEN1) in conjunction with PCNA in long-patch repair [11,15,16]. Finally, the resealing of DNA is performed by DNA ligases, predominantly ligase I and ligase III [12,17].

The poly(ADP-ribose) polymerase (PARP) family of enzymes also play a critical role in BER and its subpathway, single-strand break repair (SSBR) [18,19]. The PARP family contains at least 17 members with wide-ranging functions including cell replication and death [20,21]; however, the isoforms PARP1 and 2 are the most researched given their vital roles in DNA repair. PARP1 is formed from three major domains: a DNA-damage-sensing and -binding domain, an automodification domain, and a catalytic domain. PARP1 binds to, and is activated by, DNA breaks using its three zinc fingers; the enzyme then catalyses the addition of long, branched chains of poly(ADP-ribose) to itself and other key repair proteins. This forms a negatively charged scaffold upon which other repair proteins are recruited and repair can take place [22]. Whilst this mechanism applies to PARP1 through to PARP5 (with the exception of PARP3), other members of the PARP family only catalyse the addition of mono(ADP-ribose) and are, therefore, thought to play regulatory roles within the cell [21]. PARP-deficient cells and mice have shown greater sensitivity to DNA-damaging agents; conversely, the upregulation of PARP has been observed in some cancers and may contribute to drug resistance [20]. As discussed further in this review, the diverse roles of PARP proteins within the DDR make them an attractive target for cancer therapeutics.

Within BER, PARP forms a complex with DNA ligase III, XRCC1, and polβ and accelerates the repair pathway [20,23,24], although BER can occur independently of PARP [25]. On the other hand, PARP plays a more distinct role in SSBR, in which it first detects and binds to the single-strand break in DNA [26]. Following this, the DNA-bound PARP conducts poly(ADP-ribose) phosphorylation as previously described, whilst also interacting with XRCC1; the autoribosylated PARP enzyme then rapidly dissociates from DNA due to charge repulsion [27]. Subsequently, XRCC1 acts as a molecular scaffold for the remaining enzymatic repair proteins in SSBR, including polβ, APE1, polynucleotide kinase/phosphatase (PNKP), FEN1, and DNA ligase III [28]. This enzyme complex then processes the damaged termini, inserts new nucleotides at gaps, and ligates the damaged strand [28].

Evidence from cell line and knockout murine models demonstrates that the absence of key effector proteins within BER results in either embryonic lethality or an accumulation of mutations and hypersensitivity to DNA-damaging agents [29]. Furthermore, in humans, polymorphisms and mutations in the genes coding for these BER proteins, such as glycosylases, APE1, and XRCC1, have been associated with an increased risk of developing a range of cancers [29]. This serves to highlight the integral role of BER in repairing carcinogenic DNA lesions and is reviewed in greater detail in [29].

#### 1.1.4. Nucleotide Excision Repair (NER)

The nucleotide excision repair (NER) pathway recognises and repairs distorting single-strand damage [30,31] as may occur following ultraviolet light (UV) damage. NER can also be further classified into transcription-coupled NER (TC-NER) for actively transcribed DNA and global-genome NER (GG-NER) for nonactively transcribed DNA, with broadly similar pathways for both. Following the recognition of damage by sensor proteins [32,33], a nine-protein complex, transcription factor IIH, is recruited, which utilises its helicases XPB and XPD to unwind DNA. Incisions are then made around the lesion by the endonucleases XPG (3′ end) and XPF-ERCC1 (5′ end), generating an oligonucleotide product of 25–30 nucleotides in length [34]. Finally, DNA polymerases and ligases, namely polε acting with PCNA and ligase I (in replicating cells) and pol δ and κ in conjunction with PCNA and ligase IIIα/XRCC1 (in quiescent cells), act to fill and seal the gap [5,32,33]. The PARP enzymes also play a role in GG-NER by interacting with DNA damage-binding protein 2 (DDB2), causing chromatin remodelling to allow repair, and recruiting XPC, a key UV damage sensor [35]. Germline mutations in NER components result in xeroderma pigmentosum; affected patients possess an extremely strong predisposition to developing nonmelanoma skin cancers, stemming from a failure to repair UV-induced skin damage [36]. Moreover, these patients are also at an increased risk of internal tumours, likely due to the impaired NER of endogenously induced DNA lesions [37].

#### 1.1.5. Mismatch Repair (MMR)

The MMR pathway recognises and repairs DNA replication errors such as base–base mismatches and insertion/deletion loops (IDLs) which have escaped proofreading by DNA polymerases [38,39]. MMR is initiated by the MSH2-MSH6 (small mismatches) or MSH2-MSH3 (large mismatches or IDLs) heterodimers which recruit the MLH1-PMS2 heterodimer to clamp to the recognised lesion [40]. In conjunction with exonucleases, polymerases, and ligases, this ternary complex facilitates the excision and reforming of DNA using the other strand as a template [6,41]. Defective MMR, typically due to germline or somatic mutations in MSH2 and MLH1, impairs the repair of IDLs in microsatellite DNA and promotes genomic instability; such germline mutations have been shown to cause hereditary nonpolyposis colorectal cancer (HNPCC) [42].

#### 1.1.6. Nonhomologous End Joining (NHEJ)

Double-strand breaks (DSBs) may occur as a result of IR, ROS, stalled replication forks, or certain chemotherapy agents, and are considered the most cytotoxic DNA lesion [43]. DSBs are either repaired by the more error-prone and mutagenic nonhomologous end joining (NHEJ) or by the higher fidelity homologous recombination (HR) [44,45,46]. In summary, NHEJ begins with the recognition of DNA damage by the Ku protein which, in association with the DNA-dependent protein kinase catalytic subunit (DNA-PKcs), forms a DNA-PK complex. This heterodimer then recruits: XRCC4 to act as a scaffold for other effector proteins, endonucleases to process the damaged ends (in more severe damage), DNA polymerases λ and μ to insert new nucleotides where required, and DNA ligase IV to reseal damaged DNA [47]. Notably, both polymerases λ and μ can insert nucleotides in a template-independent manner (although more commonly performed by polμ), increasing the error rate of the pathway [45,46]. The defective function of the NHEJ pathway impairs the repair of DSB and, therefore, results in increased sensitivity to IR [48].

#### 1.1.7. Homologous Recombination (HR)

Conversely, the HR pathway, as its name suggests, utilises a homologous template DNA strand for the high-fidelity repair of DSBs and DNA interstrand crosslinks (ICLs). Although more accurate than NHEJ, HR is generally preferred for more complex DSBs, or those occurring during the replication or S or G2 phases of the cell cycle, given the availability of a template strand [49]. In eukaryotes, HR begins with the binding and resection of DNA at the DSB by the MRN complex (Mre11-Rad50-Nbs1) [50], facilitated by BRCA1, forming single-strand DNA which is subsequently coated by Replication Protein A (RPA). RPA is then replaced by RAD51, mediated by BRCA2. The RAD51-bound DNA searches for and invades the homologous sequence on the sister chromatid [6], again promoted by BRCA1 [51]. A range of DNA polymerases [52], with a possible preference towards polδ [53], then repair the break using the sister strand before dissociating and ligating the new ends. These final steps can occur via synthesis-dependent strand annealing (SDSA) or the creation of Holliday junctions [54], both of which are reviewed in more detail elsewhere [51,55].

ICL repair is considered a substrate of both the NER and HR pathways, utilising similar effector proteins such as XPG, XPF-ERCC1, BRCA1/2, RAD51, and RPA, in conjunction with the Fanconi Anaemia complex, Bloom’s syndrome complex, polν, and ataxia telangiectasia and Rad3-related protein (ATR). Whilst ICL repair is reviewed in detail elsewhere [56,57], it is relevant to note that platinum agents, which are often used in the management of advanced ovarian cancer, primarily act through generating ICLs. As a result, the upregulation of the ICL repair pathway may confer resistance or reduced responsiveness to platinum agents and highlights a potential therapeutic target [58].

## 2. DNA Repair and Cancer

Failure to repair these DNA lesions results in mutations, which in turn promotes neoplasia and carcinogenesis. As discussed, germline mutations and polymorphisms in DDR genes are identified causes of hereditary cancer syndromes such as HNPCC and can predispose to the development of multiple other tumours. For instance, germline mutations in the MMR proteins also increase the cumulative lifetime risk of ovarian cancer [59]. Furthermore, tumours harbouring mutations in DNA repair pathways are inherently more mutagenic. Due to selection pressures, mutations in oncogenes and tumour suppressor genes are more conducive to survival and, hence, more prevalent in these tumours, in accordance with the “mutator phenotype” [60]. Consequently, these tumours are associated with a more aggressive phenotype and poorer prognosis [61,62]. A study of ovarian cancers found that the loss of TP53, a tumour suppressor gene which has direct and indirect roles within the DDR [63], is an early event which is then followed by impairments in HR and, finally, widespread genomic instability [64]. Ovarian cancers with these mutations are typically more aggressive and of a higher grade [59].

On the other hand, the upregulation of particular repair pathways within tumours may promote resistance to DNA-damaging therapeutic modalities such as chemotherapy and radiotherapy [62]. For example, higher expression of XRCC1 (involved in BER and NER as described) is associated with platinum resistance and inferior outcomes in ovarian cancers [65]. Pharmacological inhibition of the DDR may, therefore, sensitise tumours to these treatment modalities, although such combinations carry a greater risk of systemic toxicity [66,67,68,69].

## 3. Limitations of Conventional Chemotherapy

Whilst EOCs often contain alterations in DDR pathways [70,71], they are considered a chemotherapy-sensitive malignancy with high objective response rates (in excess of 70%) using platinum-based combination regimens in treatment-naive patients. However, the development of acquired platinum resistance remains a common clinical problem and leads to a “therapeutic ceiling” with conventional chemotherapy [72]. Clinical trial data of patients with stage Ic to IV ovarian cancer treated with platinum and taxane combination chemotherapy suggest that 2-year overall survival is only around 66% [73]. For those with stage III and IV disease, the median time to radiologic progression after initial surgery and chemotherapy is as short as 12–18 months and the likelihood of 5-year overall survival is less than 35% [74]. Chemotherapy which includes a taxane agent, such as paclitaxel, is associated with a small but real (approximately 15%) risk of disabling long-term (more than 6 months) peripheral neuropathy with attendant deterioration in quality of life [75]. Furthermore, the delivery of carboplatin can be challenging given the not-uncommon risk of hypersensitivity reactions [76]. The ability of elderly, frail, or multiply comorbid patients to tolerate and benefit from combination chemotherapy is often questionable, and these patients comprise a substantial minority of those diagnosed with the condition. In those with disease relapse or progression after initial curative-intent treatment, or those with stage IV disease not amenable to any surgery, the development of platinum resistance portends a guarded prognosis, although a modest extension of survival can be achieved with second-line chemotherapy such as liposomal doxorubicin, gemcitabine, or topotecan [77]. Beyond conventional chemotherapy, antiangiogenic treatment, such as bevacizumab, has a limited additional benefit.

## 4. BRCA/HRD Mutations in Ovarian Cancer

To improve outcomes and minimize systemic treatment-related toxicity, precision oncology strategies exploiting “synthetic lethality” to treat advanced ovarian cancer have been developed. Synthetic lethality is the situation in which a loss of one of two critical genes does not cause cell death, whilst a loss of both results in cell death [78]. One such example is the synthetically lethal interaction between PARP1 and BRCA; the pharmacological inhibition of PARP1 by PARP inhibitors (PARPis) is thought to “trap” the enzyme at SSBs. The failure of SSBR results in a DSB; in BRCA-proficient cells, these lesions are repaired, whilst in BRCA-mutated tumour cells, the accumulation of unrepaired DSBs causes selective cell death [79]. However, new evidence challenges this conventional model, suggesting synthetic lethality arises from the accumulation of replication gaps in BRCA-deficient cells treated with PARPis [80]. Furthermore, tumours with deficiencies in other components of the HR pathway (HR-deficient or HRD) are said to possess “BRCAness” and, likely by the same mechanism, also demonstrate synthetic lethality with PARPis [79,81].

## 5. BRCA Deficiency

BRCA1 and BRCA2 are tumour suppressor genes that produce proteins with vital roles in DNA repair. BRCA1 has many well-studied interactions with different proteins and functions in various DDR pathways, including apoptosis and cellular checkpoint activation. Following DNA damage, BRCA1 relocates and is recruited to the site of DNA damage via molecules with signalling and mediating effects such as ATR, ATM, and H2AX. The histone ubiquitination function of E3 ubiquitin ligase RNF8 and 3 ubiquitin conjugase Ubc13 is part of the recruitment process which is facilitated by a complex of BRCA-associated molecules such as RAP80 [82]. The primary role of BRCA1 and BRCA2 in DNA repair is in promoting and conducting HR repair, as described above. Mutations of these genes, therefore, impairs the HR repair of DSBs. As such, DSB repair will be mediated by the error-prone, template-independent NHEJ pathway, leading to an accumulation of additional mutations and, thus, chromosomal instability. Individuals with a germline BRCA1 mutation carry a 39–46% risk of developing ovarian cancer [83]. This risk is much lower (11–18%) for germline BRCA 2 mutations [83]. The presence of a BRCA1 mutation has been associated with longer overall survival compared to BRCA wild-type cancers [84]. Evidence suggests that BRCA1 mutation is linked to chemosensitivity and a better prognosis in patients with ovarian cancer [82]. One study of 235 ovarian cancers found that 19% harboured either germline or somatic BRCA1/2 mutations [70] and may, therefore, be amenable to PARPi treatment.

Recent clinical trials have shown that PARPis may be beneficial in other different solid tumours including breast, pancreatic, prostate, and lung cancer. The OlympiAD trial compared PARPi with standard chemotherapy for patients with metastatic breast cancer and germline BRCA mutation. The response rate was 59.9% in the PARPi group and 28.8% in the standard therapy group. PFS was significantly longer with PARPis (7 months vs. 4.2 months) [85]. Furthermore, the OlympiA trial showed that the addition of PARPis in the adjuvant setting for patients with high-risk, early breast cancer with germline BRCA mutations was associated with significantly longer survival free of invasive or distant disease compared with placebo (3 years invasive-disease-free 85.9% vs. 77.1%) [86]. The TOPARP-A trial for metastatic prostate cancer patients demonstrated that PARPis in patients with castration-resistant prostate cancer and defects in DNA repair genes led to a high response rate of 33% [87]. HR deficiency is common in non-small-cell lung cancer patients [88], but the observed improvements in PFS with olaparib maintenance monotherapy over placebo in a recent trial were not statistically significant [89]. Approximately 20–30% of pancreatic cancer patients also have HR deficiency and the POLO study showed a significantly longer median PFS in favour of PARPis when compared with placebo (7.4 months vs. 3.8 months) [90] for treating metastatic pancreatic adenocarcinoma patients with disease control after first-line platinum-containing chemotherapy, although there was no overall survival benefit with long-term follow up.

## 6. HR Deficiency (HRD)

As discussed above, HR is an essential DSB and ICL repair pathway [91], but also plays a pivotal role in DNA replication and telomerase maintenance. HR deficiency (HRD) can occur due to BRCA 1/2 gene mutations, as described. In addition, somatic mutations, germline mutations, and epigenetic modifications of other gene promoters have all been implicated in HRD [71]. Ovarian cancers resulting from these alterations have identical behaviour to those with BRCA mutations; this phenotype is termed “BRCAness” and is present in approximately 41–50% of ovarian cancers [71]. Besides ovarian cancer, HRD has been demonstrated in several tumours including breast, pancreatic, and prostate cancers [91]. The evaluation of HRD can be established by (A) the germline mutation screening of DNA from blood lymphocytes via next-generation sequencing (NGS), (B) screening for somatic mutations on DNA obtained from tumour samples, and (C) the assessment of genomic instability (genomic scarring or signature) caused by HRD. These instability signatures entail the genomic patterns of Loss of Heterozygosity (gLOH), telomeric imbalances, and large-scale transitions (i.e., chromosomal breaks through deletions, inversions, or translocation) [92]. The evaluation of these three independent DNA-based measures (LOH, telomeric allelic imbalance, and large-scale transitions) has been combined to create a validated HRD score, with a value of ≥42 being predictive of clinical benefit from PARPi therapy [92].

## 7. Therapeutic Manipulation of DNA Damage Pathways to Treat Ovarian Cancer (Table 1)

### 7.1. PARP Inhibitors as Primary Maintenance Therapy

The standard first-line treatment for newly diagnosed advanced EOC comprises a combination of platinum–taxane chemotherapy (postoperatively or perioperatively) and maximal debulking surgery. Recurrence rates following primary treatment are high at approximately 70–90% for advanced disease [93]. Until the PARP inhibitor era, concurrent and maintenance treatment with bevacizumab offered a modest survival benefit with the drawback of additional adverse events related to antiangiogenic treatment [3]. Following FDA approval in December 2018, PARP inhibitors, through the exploitation of synthetic lethality, have become key players in first-line maintenance treatment for newly diagnosed advanced EOC. The role of PARPis in the current treatment algorithm for advanced ovarian cancer is outlined in Figure 2. Their efficacy as a primary maintenance therapy is evidenced by three large-scale phase-III trials: SOLO-I (olaparib) [94,95], PRIMA (niraparib) [96], and PAOLA-I [97,98] (olaparib plus bevacizumab).

**Table 1 ijms-24-07293-t001:** Summary of pivotal trials of PARPis for advanced ovarian cancer.

Trial Name	Study Title	Author and Year Published	BRCA-Mutated or HRD Tumours Only	PARPi	Comparator	Sample Size	mPFS PARPi vs. Comparator (months)	HR (95% CI)	Other Relevant Results
**PRIMARY MAINTENANCE THERAPY**									
**SOLO-1**	Maintenance Olaparib in Patients with Newly Diagnosed Advanced Ovarian Cancer	Moore et al., 2018 [94] and Banerjee et al., 2021 [95]	Yes	Olaparib (300 mg BD)	Placebo	391	56.0 vs. 13.8	0.33 (0.25–0.43)	Rate of freedom from disease progression: 60% olaparib and 27% placebo (*p* < 0.001)
**PRIMA**	Niraparib in Patients with Newly Diagnosed Advanced Ovarian Cancer	González-Martín et al., 2019 [96]	No	Niraparib (200 mg or 300 mg OD)	Placebo	733	All: 13.8 vs. 8.2 HRD: 21.9 vs. 10.4	All: 0.62 (0.50–0.76) HRD: 0.43 (0.31–0.59)	OS: 84% in niraparib group vs. 77% in placebo group at 24 months
**PAOLA-1**	Olaparib plus Bevacizumab as First-Line Maintenance in Ovarian Cancer	Ray-Coquard et al., 2019 [97]	No	Olaparib (300 mg BD) plus bevacizumab (15 mg/kg IV 3 weekly)	Placebo plus bevacizumab (15 mg/kg IV 3 weekly)	806	All: 22.1 vs. 16.6 HRD: 28.1 vs. 16.6	All: 0.33 (0.25–0.45) HRD: 0.43 (0.28–0.66)	
**OVARIO**	OVARIO phase II trial of combination niraparib plus bevacizumab maintenance therapy in advanced ovarian cancer following first-line platinum-based chemotherapy with bevacizumab	Hardesty et al., 2022 [99]	No	Niraparib (200 or 300 mg OD) plus bevacizumab (15 mg/kg IV 3 weekly)	Nil	105	All: 19.6 HRD: 28.2 HR-proficient: 14.2		PFS rate at 18 months: All: 62% HRD: 76% HR-proficient: 56%
**ATHENA-MONO**	A Randomized, Phase III Trial to Evaluate Rucaparib Monotherapy as Maintenance Treatment in Patients With Newly Diagnosed Ovarian Cancer (ATHENA-MONO/GOG-3020/ENGOT-ov45)	Monk et al., 2022 [100]	No	Rucaparib (600 mg BD)	Placebo	538	All: 20.2 vs. 9.2 HRD: 28.7 vs. 11.3 HR-proficient: 12.1 vs. 9.1	All: 0.52 (0.40–0.68) HRD: 0.47 (0.31–0.72) HR-proficient: 0.65 (0.45–0.95)	All ORR: 48.8% in rucaparib group vs. 9.1% in placebo group HRD ORR: 58.8% in rucaparib group vs. 20% in placebo
**VELIA**	Veliparib with First-Line Chemotherapy and as Maintenance Therapy in Ovarian Cancer	Coleman et al., 2019 [101]	No	Veliparib (150 mg OD) plus chemotherapy followed by Veliparib maintenance	Chemotherapy plus placebo, chemotherapy plus veliparib followed by placebo maintenance	1140	All: 23.5 vs. 17.3 gBRCA: 34.7 vs. 22.0 HRD: 31.9 vs. 20.5	All: 0.68 (0.56–0.83) eBRCA: 0.44 (0.28–0.68) HRD: 0.57 (0.43–0.76)	ORR: 84% in veliparib-throughout group vs. 74% in the control group after six chemotherapy cycles
**RECURRENT MAINTENANCE THERAPY**									
**SOLO-2**	Olaparib tablets as maintenance therapy in patients with platinum-sensitive, relapsed ovarian cancer and a BRCA1/2 mutation (SOLO2/ENGOT-Ov21): a double-blind, randomised, placebo-controlled, phase 3 trial	Pujade-Lauraine et al., 2017 [102]	Yes	Olaparib (300 mg BD)	Placebo	295	19.1 vs. 5.5	0.30 (0.22–0.41)	24 months without disease progression rate 43.0% in olaparib group vs. 15.1% in placebo group
**NOVA**	Niraparib Maintenance Therapy in Platinum-Sensitive, Recurrent Ovarian Cancer^21^	Mirza et al., 2016 [103]	No	Niraparib (300 mg OD)	Placebo	553	gBRCA: 21.0 vs. 5.5 non-gBRCA: 9.3 vs. 3.9 HRD: 12.9 vs. 3.8	gBRCA: 0.27 (0.17–0.41) non-gBRCA: 0.45 (0.34–0.61) HRD: 0.38 (0.24–0.59)	
**ARIEL3**	Rucaparib maintenance treatment for recurrent ovarian carcinoma after response to platinum therapy (ARIEL3): a randomised, double-blind, placebo-controlled, phase 3 trial	Coleman et al., 2017 [104]	No	Rucaparib (600 mg BD)	Placebo	564	All:10.8 vs. 5.4 BRCAm:16.6 vs. 5.4 HRD: 13.6 vs. 5.4	0.36 (0.30–0.45)	
**OReO**	Maintenance olaparib rechallenge in patients with ovarian carcinoma previously treated with a PARP inhibitor (PARPi): Phase IIIb OReO/ENGOT Ov-38 trial	Pujade-Lauraine et al., 2021 [105]	No	Olaparib (300 mg BD)	Placebo	220	BRCAm: 4.3 vs. 2.8 non-BRCAm: 5.3 vs. 2.8	BRCAm: 0.57 (0.37–0.87) non-BRCAm: 0.43 (0.26–0.71)	
**MONOTHERAPY FOR RELAPSED DISEASE**									
**SOLO-3**	Olaparib Versus Nonplatinum Chemotherapy in Patients With Platinum-Sensitive Relapsed Ovarian Cancer and a Germline BRCA1/2 Mutation (SOLO3): A Randomized Phase III Trial	Penson et al., 2020 [106]	Yes	Olaparib (300 mg BD)	Physician’s choice single-agent nonplatinum chemotherapy	266	13.4 vs. 9.2	0.62 (0.43–0.91)	ORR: 72.2% for olaparib vs. 51.4% for chemotherapy
**ARIEL4**	Overall survival results from ARIEL4: A phase III study assessing rucaparib vs. chemotherapy in patients with advanced, relapsed ovarian carcinoma and a deleterious BRCA1/2 mutation	Oza et al., 2022 [107] and Kristeleit et al., 2022 [108]	Yes	Rucaparib (600 mg BD)	Chemotherapy	349	7.4 vs. 5.7	0.67 (0.52–0.86)	Median OS: rucaparib group 19.4 months vs. 25.4 months in chemotherapy group
**PARP-INHIBITOR-BASED COMBINATION STRATEGIES WITH CHEMOTHERAPY FOR RELAPSED DISEASE**									
**N/A**	Olaparib combined with chemotherapy for recurrent platinum-sensitive ovarian cancer: a randomised phase 2 trial	Oza et al., 2015 [109]	No	Olaparib (200 mg BD) plus chemotherapy followed by Olaparib maintenance monotherapy	Chemotherapy	162	12.2 vs. 9.6	0.51 (0.35–0.77)	Olaparib especially effective in BCRm: HR 0.21 (0.08–0.55)
**N/A**	Randomized Trial of Oral Cyclophosphamide and Veliparib in High-Grade Serous Ovarian, Primary Peritoneal, or Fallopian Tube Cancers, or BRCA-Mutant Ovarian Cancer	Kummar et al., 2015 [110]	No	Veliparib (60 mg OD) plus cyclophosphamide (50 mg OD)	Cyclophosphamide (50 mg OD) alone	75	2.1 vs. 2.3	NA	One complete response in each arm, three partial responses in the veliparib group, and six partial responses in the cyclophosphamide group
**ROLANDO**	Olaparib in combination with pegylated liposomal doxorubicin for platinum-resistant ovarian cancer regardless of BRCA status: a GEICO phase II trial (ROLANDO study)	Perez-Fidalgo et al., 2021 [111]	No	Olaparib (300 mg BD) plus PLD chemotherapy, followed by maintenance olaparib	Nil	31	5.8	NA	Overall disease control rate 77% (29% partial response, 48% stable disease)
**PARP INHIBITOR-BASED COMBINATION STRATEGIES WITH ANTIANGIOGENIC THERAPY FOR RELAPSED DISEASE**									
**N/A**	Overall survival and updated progression-free survival outcomes in a randomized phase II study of combination cediranib and olaparib versus olaparib in relapsed platinum-sensitive ovarian cancer	Liu et al., 2019 [112]	No	Olaparib (200 mg BD) plus cediranib (30 mg OD)	Olaparib (400 mg BD) alone	90	All: 16.5 vs. 8.2 Non-BRCA/unknown: 23.7 vs. 5.7	0.50 (0.30–0.83)	Median OS: combination arm 44.2 vs. 33.3 months in monotherapy
**NSGO-AVANOVA2/ENGOT-ov24**	Niraparib plus bevacizumab versus niraparib alone for platinum-sensitive recurrent ovarian cancer (NSGO-AVANOVA2/ENGOT-ov24): a randomised, phase 2, superiority trial	Mirza et al., 2019 [113]	No	Niraparib (300 mg OD) plus bevacizumab (15 mg/kg IV 3 weekly)	Niraparib (300 mg OD) alone	97	11.9 vs. 5.5	0.35 (0.21–0.57)	ORR: combination 60% vs. 27% niraparib alone

SOLO-I [94,95] established the role of PARP inhibitors as primary maintenance therapy. The trial enrolled 391 patients with newly diagnosed BRCA-mutant stage 3/4 ovarian cancer with platinum-sensitive disease. Patients were randomised to olaparib or placebo for two years or until treatment-limiting toxicity or progression. Initial analysis at 41 months showed a 70% lower risk of progression or death with olaparib. At 5 years, 48% of patients in the treatment arm remained free of progression or recurrence compared to 21% with placebo. The median progression-free survival (PFS) with olaparib was 56 months versus 13.8 months with placebo (HR 0.33 95% CI 0.25–0.43). At 7-year follow-up, the median OS was 75 months with placebo and had not yet been reached with olaparib (HR 0.55 95% CI 0.40–0.76). Generally, olaparib was largely well tolerated, with the most frequent grade-3/4 toxicity being anaemia and thrombocytopenia. There was a 1% incidence of acute myeloid leukaemia (n = 3/260) in the olaparib arm at initial analysis with no further cases at 5-year follow-up. The health-related quality of life was equivalent in both arms.

The PRIMA trial [96] investigated maintenance PARP inhibitors in genomically unstratified advanced serous or endometrioid ovarian cancer patients who were platinum-sensitive. Patients in the PRIMA trial had more advanced disease than those in SOLO-1, with a lower proportion of optimally debulked stage 3 patients (0.4% vs. 44%). Patients were randomised to up to 3 years of niraparib or placebo. The coprimary endpoints of PRIMA were to determine the PFS of the entire population as well as the PFS of patients deemed HR-deficient based on their BRCA status or Myriad score. There was a 57% reduction in relapse or death with niraparib in the HR-deficient population (HR 0.43 95% CI, 0.21–0.59), with a median PFS of 21.9 months versus 10.4 in placebo. Crucially, the trial also found a clinically significant hazard ratio of 0.62 (95% CI, 0.50–0.76) in the overall population, with a median PFS of 13.8 months with niraparib versus 8.2 months with placebo. These data led to the FDA approval of niraparib as primary maintenance for all advanced-stage platinum-sensitive ovarian cancer.

Antiangiogenics such as bevacizumab may have a synergistic effect with PARP inhibitors through the hypoxia-mediated downregulation of homologous recombination repair [114]. The PAOLA-I trial [97,98] studied the effect of combined olaparib and bevacizumab in the maintenance setting for all patients with newly diagnosed, advanced-stage, platinum-responsive EOC regardless of BRCA/HRD status. The patients were randomised to receive a 2-year course of maintenance bevacizumab, given concurrently with either olaparib or placebo. The combination treatment was well tolerated, with no significant deterioration in quality-of-life analyses. A planned subgroup analysis of HRD patients inclusive of BRCA mutants found a median PFS of 37.2 months in the combined group compared to 17.7 months with bevacizumab alone (HR 0.33 95% CI, 0.25–0.45). Similar results were observed in HRD patients excluding BRCA mutants (HR 0.43 95% CI, 0.28–0.66). Unlike the PRIMA trial, there was no survival benefit in the HR-proficient/unknown cohort, thus excluding these patients from the use of combination olaparib–bevacizumab as maintenance treatment. Since the trial did not include a single-agent olaparib arm, it is not possible to determine the extent of any synergism between olaparib and bevacizumab. Other PARP inhibitors have been similarly investigated for use with antiangiogenic agents. The single-arm, phase-II OVARIO-trial [99] assessed the safety and efficacy of combination niraparib and bevacizumab as first-line maintenance for newly diagnosed advanced-stage ovarian cancer. Analysis at 28.7 months found a median PFS of 28.3 months for HR-deficient patients compared to 14.2 months for those who were HR-proficient.

Additional PARP inhibitors are on the horizon for approval in the first line-maintenance setting. Among these is rucaparib, which was evaluated in the phase-III ATHENA-MONO trial [100]. Five hundred and thirty-eight all-comers with advanced high-grade EOC with evidence of platinum response were randomised 4:1 to rucaparib alone or placebo. The median PFS in the HR-deficient cohort was 28.7 months with rucaparib versus 11.3 months with placebo (HR 0.47; 95% CI, 0.31 to 0.72). As with the PRIMA trial, a survival benefit was sustained in the overall intention-to-treat population, who had a median PFS of 20.2 months with rucaparib compared to 9.2 months with placebo (HR, 0.52; 95% CI, 0.40 to 0.68). Finally, the VELIA phase-III study [101] considered the use of PARP inhibitors as a synergistic adjunct to platinum-based chemotherapy. This was made possible by the favourable haematological toxicity profile of veliparib due to its relatively weak PARP-trapping capacity [79]. Participants with advanced EOC were randomised to either chemotherapy alone (control arm), chemotherapy plus veliparib followed by maintenance placebo, or concurrent chemotherapy and veliparib followed by veliparib maintenance. This study was limited by dose reductions, treatment interruption, and discontinuation; however, the results showed an improvement in PFS in the veliparib-throughout arm that was most pronounced in the BRCA-mutated cohort. The trial would have benefited from a further arm comprising chemotherapy alone followed by veliparib maintenance, as this would allow a comparison of relative contributions to PFS and the benefit (if any) of adding veliparib during induction.

### 7.2. PARP Inhibitors as Recurrent Maintenance Therapy

PARP inhibitors initially gained regulatory approval for use in the maintenance setting for relapsed platinum-sensitive ovarian cancers. The pivotal phase-III trials that led to approvals include SOLO-2 (olaparib) [102], NOVA (niraparib) [103], and ARIEL3 (rucaparib) [104]. Patients recruited into these studies had a background of recurrent high-grade ovarian cancer who had previously received two or more lines of platinum-based chemotherapy and had achieved complete or partial response to the most recent regimen. Overall, the results from all trials were consistent with a survival benefit in the general population, albeit to varying extents based on biomarker status. It is important to note that all patients were PARPi-naïve as the three trials took place prior to the introduction of PARP inhibitors as primary maintenance. A more recent phase-III trial has, however, shown promising results with respect to PFS following PARP rechallenge in extensively pretreated ovarian cancer patients [105].

The SOLO-2 [102] trial randomised 295 eligible patients with germline or somatic BRCA mutations to receive olaparib or placebo until disease progression. Patients in the olaparib group had a significantly longer median PFS of 19.1 months compared to 5.5 months with placebo (HR 0.30; 95% CI, 0.22–0.41). Over 40% of patients in SOLO-2 had received a minimum of three prior lines of chemotherapy. This was reflected in the rate of therapy-related leukaemias (myelodysplasia and acute myeloid leukaemia), which are known adverse events of extensive prior chemotherapy, particularly with platinum agents. The rate of MDS/AML was 8% in the olaparib arm compared to 4% in the placebo arm, which might impact overall survival. The median OS in the olaparib arm was 51.7 months compared to 38.8 months with placebo (HR 0.74; 95% CI, 0.54–1.00 *p* = 0.054), despite 38% of placebo patients crossing over to olaparib. This was, however, not statistically significant as the study was not powered to determine OS. The side effects of olaparib were generally low-grade and well-tolerated, with no detriment to quality of life.

To investigate the use of maintenance PARP inhibitors outside the BRCA mutant cohort, the NOVA study [103] recruited eligible patients with recurrent ovarian cancer regardless of BRCA and HRD status; 553 patients were randomised to receive either niraparib or placebo. The median PFS in patients with the germline BRCA mutation was 21 months with niraparib compared to 5.5 months with placebo (HR 0.27; 95% CI, 0.17–0.41). Patients with no BRCA mutation still had a median PFS of 9.3 months with niraparib versus 3.9 months with placebo (HR 0.45; 95% CI, 0.34–0.61). The HRD subgroup of this population had a median PFS of 12.9 months and 3.8 months with niraparib and placebo, respectively (HR, 0.38; 95% CI, 0.24 to 0.59). The data, therefore, demonstrate an improved median PFS in all cohorts, irrespective of biomarker status. The side-effect profile of niraparib was comparable to other PARP inhibitors, with the most frequent grade-3 and -4 toxicities being anaemia and thrombocytopenia. Therapy-related leukaemias were observed in both the niraparib cohort (n = 5/367) and the placebo cohort (n = 2/186). As with SOLO-2, NOVA did not demonstrate a statistically significant difference in median OS. The patient-reported quality-of-life outcomes were similar in both arms.

The efficacy of rucaparib was evaluated in the ARIEL3 trial [104], with 564 patients randomised to rucaparib or placebo, stratified based on BRCA status and HRD status (as determined by the extent of gLOH). The median PFS in the BRCA-mutant cohort was 16.6 months with rucaparib compared to 5.4 months with placebo (HR 0.23, 95% CI 0.16–0.34, *p* < 0.0001). HR-deficient patients (comprising both BRCA mutant and BRCA wild-type patients with more than a 16% loss of heterozygosity) had a median PFS of 13.6 months with rucaparib compared to 5.4 months with placebo (HR 0.32, 95% CI 0.24–0.42; *p* < 0.0001). Similar results were seen in the intention-to-treat population, with a median-PFS of 10.8 months with rucaparib versus 5.4 months with placebo (HR 0.36, 95% CI 0.30–0.45; *p* < 0.0001). The study reported PFS benefits in all the analysed subgroups, including in patients with a low percentage loss of heterozygosity (LOH). This indicates a need to review the prespecified cut-off of 16% or more for high genomic LOH; however, it also reinforces the possibility of PARPis having a mechanism of action beyond blocking DNA damage repair. Currently, LOH is best used as a predictive, rather than prognostic, biomarker.

### 7.3. PARP Inhibitor Monotherapy for Relapsed Advanced Ovarian Cancer

PARPis have been investigated as an alternative to traditional chemotherapy for relapsed platinum-sensitive ovarian cancer that had failed a minimum of two previous lines of treatment. The past year has seen the withdrawal all FDA-approved PARP inhibitors for heavily pretreated ovarian cancer (olaparib, niraparib, and rucaparib). This change was driven by long-term survival data from SOLO3 (olaparib) [106,115] and ARIEL-4 (rucaparib) [107,108], both of which are large phase-III trials that failed to demonstrate an overall survival benefit with PARP inhibitors when compared to chemotherapy. Based on these data, GSK voluntarily withdrew niraparib as a fourth-line treatment option for advanced ovarian cancer.

SOLO3 [106,115] compared single-agent olaparib with physicians’ choice of nonplatinum chemotherapy (PLD/gemcitabine/topotecan) in platinum-sensitive, BRCA-mutated ovarian cancer patients who had received at least two prior lines of platinum-based treatment. Platinum sensitivity was defined by a minimum of six months of progression-free survival following the last platinum-based regimen. The study enrolled 266 patients with 2:1 randomisation to receive either olaparib or chemotherapy. Despite promising initial data [106], the final analysis [115] failed to show a statistically significant survival benefit with olaparib. PFS2 was found to be 23.6 months with olaparib compared to 19.6 months with chemotherapy (HR 0.80; 95% CI, 0.56–1.15; *p* = 0.229). The median OS was 34.9 months with olaparib versus 32.9 months with chemotherapy (HR 1.07; 95% CI, 0.76–1.49). In addition to this, olaparib was associated with increased side effects, resulting in a recent subgroup analysis raising concerns of “survival detriment” in patients who had received three or more lines of prior chemotherapy. Olaparib was, therefore, withdrawn as a fourth-line treatment option as of August 2022.

ARIEL4 [107,108] reported similarly underwhelming results for rucaparib as third-line treatment. Patients were randomised to receive either rucaparib or platinum-sensitivity-guided chemotherapy. Paclitaxel was the treatment of choice for platinum-resistant or partially platinum-sensitive patients. Fully platinum-sensitive patients received either single-agent- or doublet-platinum-based chemotherapy. Patients from the chemotherapy arm were able to crossover to the rucaparib arm following disease progression (n = 80/116). The final analysis found a lower OS of 19.4 months with rucaparib compared to 25.4 months with chemotherapy (HR 1.31; 95% CI, 1.00 to 1.73; *p* = 0.0507). PFS2 was not significantly different between the two groups. Rucaparib was, therefore, withdrawn as a third-line treatment option as of June 2022.

### 7.4. PARP-Inhibitor-Based Combination Strategies for Relapsed Advanced Ovarian Cancer

#### 7.4.1. Chemotherapy

The combination of PARP inhibitors with conventional chemotherapy is complicated by overlapping toxicity profiles, in particular myelosuppression. The extent of any synergism and the nature of combined toxicity are predominantly determined by the class of cytotoxic in question. Furthermore, since the cytotoxic effect of PARP inhibitors stems from their individual PARP-trapping effect [116], less potent agents (such as veliparib) might be more suitable for combination treatment. DNA-damaging cytotoxics, such as platinum, and alkylating agents, such as cyclophosphamide and temozolomide, are of particular interest in conjunction with PARP inhibitors as their effect can be amplified through the PARP-mediated disruption of DNA repair. Other classes have also been investigated for concurrent use with PARP inhibitors, including pegylated liposomal doxorubicin (PLD) and mitomycin C (MMC).

Given the long-established role of platinum agents in ovarian cancer, their combination with PARP inhibitors was among the first to be studied. One international phase-II trial [109] randomised 163 patients with recurrent platinum-sensitive ovarian cancer to receive either chemotherapy alone or combination olaparib with carboplatin and paclitaxel with subsequent olaparib maintenance. Despite patients receiving lower doses of olaparib during the combination phase of treatment, the study showed an improved median PFS with combined therapy (12.2 months versus 9.6 months). There was, however, an increased incidence of grade-3–4 neutropenia with combination treatment (43% vs. 35%), although the incidence of other haematological toxicities was comparable between the two groups. The discontinuation rates were also broadly similar.

The long-term, low-dose (metronomic) administration of cyclophosphamide has been a viable treatment strategy for heavily pretreated ovarian cancer patients with limited tolerance to toxicity. A phase-II trial [110] comparing combination veliparib and cyclophosphamide with veliparib alone for recurrent BRCA-mutated ovarian cancer failed to show a significantly different survival benefit. The combination treatment was, however, well tolerated and incurred minimal interruptions to PARP inhibition. It is possible that the lack of survival benefit in the combination arm was due to a lower dose of veliparib in the study compared to standard single-agent doses.

Pegylated liposomal doxorubicin is yet another alternative for heavily pretreated ovarian cancer that can either be used in combination with platinum agents or as a single agent for platinum-resistant cancer. The ROLANDO study [111] investigated the use of combination PLD-olaparib for platinum-resistant ovarian cancer regardless of BRCA status. The results were consistent with an impressive disease control rate of 77% (29% partial response and 48% stable disease); however, 74% of patients incurred grade-3+ toxicities, resulting in dose delays and reductions. Serious adverse events were far less frequent with a PLD dose of 30 mg/m^2^ (21%) compared to 40 mg/m^2^ (47%). Finally, there are multiple phase-I/II trials comparing various combinations of PARP inhibitors with chemotherapy. Examples include veliparib with MMC (NCT01017640), veliparib with topotecan (NCT01012817), and veliparib with temozolomide (NCT01113957). A summary of these studies is shown in Table 1.

#### 7.4.2. Antiangiogenics

The hypoxic tumour microenvironment brought about by antiangiogenic agents (such as bevacizumab and cediranib) is thought to downregulate the expression of key gene products involved in homologous recombination repair (BRCA1/2 and RAD51) [114]. Antiangiogenics can, therefore, uniquely induce contextual HR deficiency in HR-proficient tumours. Multiple phase-II trials have investigated the survival outcomes for combination treatment. A study by Liu et al. [112] randomised patients with relapsed, platinum-sensitive ovarian cancer to receive either olaparib alone or combination olaparib with cediranib. The median PFS was 16.5 months in the combination arm compared to 8.2 months with olaparib alone (HR 0.50; 95% CI 0.30–0.83; *p* = 0.006). Post hoc analyses showed that the improvement in PFS was particularly pronounced in the BRCA wild-type/unknown population. A similar trial in the same setting compared niraparib and bevacizumab with niraparib alone and again found improvements in the PFS and objective response rates [113]. Phase-III trials are currently ongoing, such as the ICON 9 trial [117] comparing combination cediranib/olaparib with olaparib alone in the maintenance setting for advanced, platinum-sensitive, relapsed ovarian cancer. 

## 8. Interplay between DNA Damage and Antitumour Immune Responses

In the last decade, the therapeutic manipulation of T-cell immune checkpoints with monoclonal antibodies has revolutionised the management of and outlook for many advanced solid malignancies that were historically considered “immunologically cold”, such as non-small-cell lung cancer, urothelial carcinoma, and squamous head and neck cancer. However, ovarian cancer exhibits a very modest response to anti-PD-1/PD-L1 monotherapy, with responses in approximately 10–15% of chemotherapy-refractory patients and a median progression-free survival of 3–4 months [118,119]. 

Defective DNA damage repair may promote tumoural genomic instability and lead to a high tumour mutational burden, which in turn maximises neoepitope supply for potential T-cell recognition, and this is associated with a therapeutic response to immune checkpoint inhibitors. The level of CD3 and CD8 T-lymphocyte infiltration in high-grade serous ovarian carcinomas is higher, the neoantigen load is greater, and tumour-related immune cells show higher PD1 and PD-L1 expression in homologous-recombination-deficient versus -proficient tumours [120]. PARP inhibitors have been shown to promote antitumour activity by upregulating PD-L1 expression in animal models [121] and cancer cell lines [122] of BRCA mutant serous ovarian cancer and through T-cell recruitment [123].

From the clinical perspective, there is emerging early-phase data that indicate that the combination of a PARP inhibitor and anti-PD-1 therapy may have some efficacy for chemotherapy-refractory advanced epithelial ovarian cancer, with an objective radiologic response rate of 18% and disease control rate of 65% [124] and 6-month and 12-month progression-free-survival probabilities of 31% and 12%, respectively. In the relapsed-but-platinum-sensitive setting, a chemotherapy-free regimen of olaparib with concurrent durvalumab achieved a response rate of 72%, and complete responses were also seen with a 2-year overall survival of 87% [125]. Similar promising results with this combination were seen in an additional phase-II trial with a predominantly platinum-resistant population [126].

## 9. Conclusions

Efficient DNA repair is fundamental to the maintenance of genomic integrity, which is critical for cellular homeostasis. Suboptimal DNA repair will predispose to mutation accumulation and eventually increase the risk of cancer development. However, such DNA repair deficiency states can also be exploited for precision oncology through synthetic lethality. The clinical impact of PARP inhibitors in *BRCA*-germline-deficient or platinum-sensitive ovarian cancer and other tumours (such as breast, pancreatic, and prostate cancers) has been established. However, the clinical benefit is not sustained, and the eventual development of resistance is an emerging clinical problem. Understanding the mechanisms of acquired resistance remains a key priority for investigation and combinatorial approaches; for example, the use of inhibitors of cell cycle checkpoints (such as the ATR kinase) concurrently with PARPis to delay the development of resistance is currently being explored in early-phase trials (e.g., NCT03682289). Enhancing the potency of existing PARPis is also an area of interest, with approaches such as hydrophobic tagging or proteolysis-targeting chimeras. Gaining a better understanding of clinical resistance and the development of alternative synthetic lethality approaches remains a high priority in ovarian cancer precision oncology therapeutics.

## Figures and Tables

**Figure 1 ijms-24-07293-f001:**
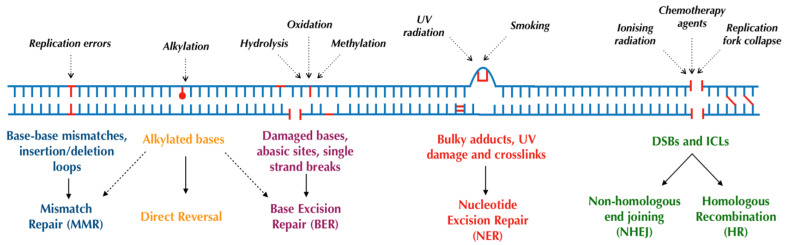
Major DNA repair pathways in mammalian cells.

**Figure 2 ijms-24-07293-f002:**
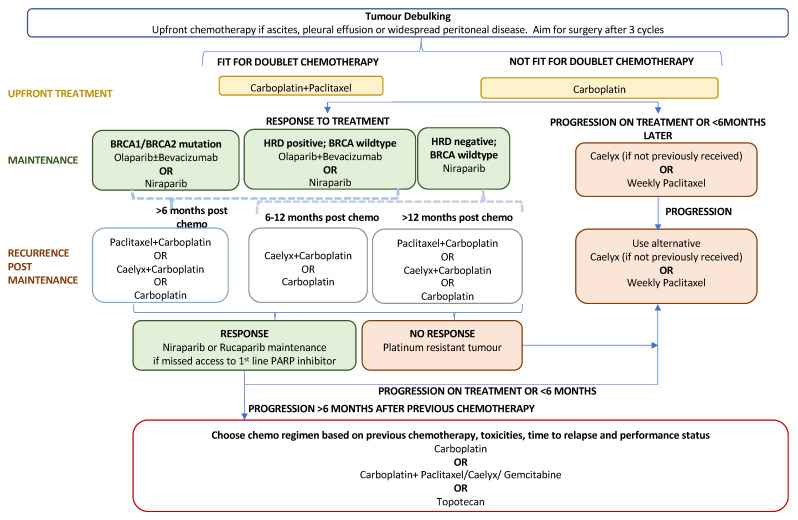
Treatment algorithm for advanced epithelial ovarian cancer.

## Data Availability

Not applicable.

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
