# Peer review of "The Current Status of DNA-Repair-Directed Precision Oncology Strategies in Epithelial Ovarian Cancers"

_ijms, 2023, doi:10.3390/ijms24087293_

Round 1
Reviewer 1 Report
Madhusudan et.al wrote a wonderful review on Current status of DNA repair directed precision oncology strategies in epithelial ovarian cancers. I have few suggestions as follows to improve this reviews.
1) Author should briefly discuss the cancer and other disease associated with DNA repair defects like BER,NER. They should cite some recent review and paper to support it.
2) ICL repair is also substrate of NER. Authors have not discussed it.
3) Since this review mainly revolves around PARP and BRCA proteins as potential targets, author should briefly describe the PARP proteins as well, their role and functions in repair and cancer.
4) Section 2 is general and vague. Author should improve it with some examples.
5) It would be interesting to discuss the similar strategy (BRCAness and PARPi) used in different cancer treatment other than EOC.
6) Author should also include the limitation and scope of this approach in conclusions.
Author Response
REVIEWER 1:
Madhusudan et.al wrote a wonderful review on Current status of DNA repair directed precision oncology strategies in epithelial ovarian cancers.
Author response: Thank you.
I have few suggestions as follows to improve this reviews.
1) Author should briefly discuss the cancer and other disease associated with DNA repair defects like BER,NER. They should cite some recent review and paper to support it.
Author response: As suggested by the reviewer, these have now been discussed with new references in the revised manuscript as below. The relevance of MMR and NHEJ repair to cancer has also been discussed (lines 113-114 and 124-125 respectively).
Lines 85-90: Evidence from cell line and knockout murine models demonstrate that the absence of key effector proteins within BER results in either embryonic lethality or an accumulation of mutations and hypersensitivity to DNA-damaging agents [26]. Furthermore, in humans, polymorphisms and mutations in the genes coding for these BER proteins, such as glycosylases, APE1 and XRCC1, have been associated with an increased risk of developing a range of cancers [26]. This serves to highlight the integral role of BER in repairing carcinogenic DNA lesions and is reviewed in greater detail in [26].
Lines 102-105: Germline mutations in NER components results in xeroderma pigmentosum; affected patients possess an extremely strong predisposition to developing non-melanoma skin cancers, stemming from a failure to repair UV-induced skin damage [33]. Moreover, these patients are also at an increased risk of internal tumours, likely due to impaired NER of endogenously induced DNA lesions [34].
2) ICL repair is also substrate of NER. Authors have not discussed it.
Author response: As suggested by the reviewer, we have included ICL repair in the revised manuscript as below (lines 138-139):
“ICL repair is considered a substrate of both the NER and HR pathways, utilising similar effector proteins such as XPG, XPF-ERCC1, BRCA1/2, RAD51 and RPA, in conjunction with the Fanconi Anaemia complex, Bloom’s syndrome complex, polν and ataxia telangiectasia and Rad3-related protein (ATR).”
3) Since this review mainly revolves around PARP and BRCA proteins as potential targets, author should briefly describe the PARP proteins as well, their role and functions in repair and cancer.
Author response: As suggested by the reviewer, further discussion of the structure and function of PARP family of proteins and their relevance to DNA repair, cell functioning and cancer has been added with additional references as below (lines 66-74):
“PARP1 is formed from three major domains: a DNA-damage sensing and binding domain, an automodification domain and a catalytic domain. PARP1 binds to, and is activated by, DNA breaks using its three zinc fingers; the enzyme then catalyses the addition of long, branched chains of poly(ADP-ribose) to itself and other key repair proteins. This forms a negatively charged scaffold upon which other repair proteins are recruited and repair can take place [19]. Whilst this mechanism applies to PARP1 through to PARP5 (with the exception of PARP3), other members of the PARP family only catalyse the addition of mono(ADP-ribose) and are therefore thought to play regulatory roles within the cell [18]. PARP-deficient cells and mice have shown greater sensitivity to DNA-damaging agents; conversely, upregulation of PARP has been observed in some cancers and may contribute to drug resistance [17]. As discussed further in this review, the diverse roles of PARP proteins within the DDR make them an attractive target for cancer therapeutics.”
4) Section 2 is general and vague. Author should improve it with some examples.
Author response: As suggested by the reviewer, Section 2 has been expanded to discuss the relationship between DNA repair and cancer/cancer therapeutics. This has now been illustrated with examples and additional references as below (lines 146-161):
“Failure to repair these DNA lesions results in mutations which in turn promotes neoplasia and carcinogenesis. As discussed, germline mutations and polymorphisms in DDR genes are identified causes of hereditary cancer syndromes such as HNPCC and can predispose to the development of multiple other tumours. For instance, germline mutations in the MMR proteins also increase the cumulative lifetime risk of ovarian cancer [56]. Furthermore, tumours harbouring mutations in DNA repair pathways are inherently more mutagenic. Due to selection pressures, mutations in oncogenes and tumour suppressor genes are more conducive to survival and hence more prevalent in these tumours, in accordance with the ‘mutator phenotype’ [57]. Consequently, these tumours are associated with a more aggressive phenotype and poorer prognosis [58,59]. A study of ovarian cancers found that loss of TP53, a tumour suppressor gene which has direct and indirect roles within the DDR [60], is an early event which is then followed by impairments in HR and finally widespread genomic instability [61]. Ovarian cancers with these mutations are typically more aggressive and of a higher grade [56].”
“On the other hand, up-regulation of particular repair pathways within tumours may promote resistance to DNA-damaging therapeutic modalities such as chemotherapy and radiotherapy [59]. For example, higher expression of XRCC1 (involved in BER and NER as described) is associated with platinum resistance and inferior outcomes in ovarian cancers [62]. Pharmacological inhibition of the DDR may therefore sensitise tumours to these treatment modalities, although such combinations carry greater risk of systemic toxicity [63-66]. “
5) It would be interesting to discuss the similar strategy (BRCAness and PARPi) used in different cancer treatment other than EOC.
Author response: As suggested by the reviewer, we have now added to Section 5 a paragraph outlining the evidence for PARPi in other BRCA-mutated and HRD tumours including breast, pancreatic prostate and lung, with supporting evidence from recent clinical trials (lines 211-224):
“Recent clinical trials have shown PARPi may be beneficial in other different solid tumours including breast, pancreatic, prostate and lung cancer. The OlympiAD trial compared PARPi with standard chemotherapy for patients with metastatic breast cancer and germline BRCA mutation. The response rate was 59.9% in the PARPi group and 28.8% in the standard therapy group. PFS was significantly longer with PARPi (7 months vs 4.2 months) [82]. Furthermore, the OlympiA trial showed that the addition of PARPi in the adjuvant setting for patients with high risk, early breast cancer with germline BRCA mutations, was associated with significant longer survival free of invasive or distant disease compared with placebo (3 years invasive disease free 85.9% vs 77.1%) [83]. The TOPARP-A trial for metastatic prostate cancer patients demonstrated PARPi in patients with castration resistant prostate cancer and defects in DNA repair genes led to a high response rate of 33% [84]. HR deficiency common in non-small cell lung cancer patients [85] but the observed improvements in PFS with olaparib maintenance monotherapy over placebo in a recent trial were not statistically significant [86]. Approximately 20-30% of the pancreatic cancer patients also have HR deficiency and the POLO study showed significant longer median PFS in favour of PARPi when compared with placebo (7.4 months vs 3.8 months) [87] for treating metastatic pancreatic adenocarcinoma patients with disease control after first line platinum-containing chemotherapy, although there was no overall survival benefit with long term follow up.”
6) Author should also include the limitation and scope of this approach in conclusions
Author response: As suggested by the reviewer, this has now been mentioned in the revised conclusion with suggestions for potential areas of future research as below (lines 470-474):
Understanding the mechanisms of acquired resistance remains a key priority for investigation and combinatorial approaches such as the use of inhibitors of cell-cycle checkpoints (such as the ATR kinase) concurrently with PARPi to delay the development of resistance need are currently being explored in early-phase trials (e.g., NCT03682289). Enhancing the potency of existing PARPi is also an area of interest with approaches such as hydrophobic tagging or proteolysis-targeting chimeras.
Reviewer 2 Report
Authors described current status of DNA repair directed precision oncology strategies in epithelial ovarian cancers as a Review. There are many references beyond those listed by the authors, but this is a review and it is up to the authors to decide what to add. Therefore, we believe that the content is mostly acceptable. However, the following points could be revised to make the review more comprehensible to readers.
1. To help readers understand, I strongly recommend adding a single, easy-to-understand diagram of the "strategy" using an entire page, as authors described "strategy" in the title.
2. There are very many abbreviations. A list of abbreviations should be included for readers.
3. In Abstract,
"PARP" should be changed to "poly(ADP-ribose) polymerase (PARP)".
4. In line 124, "polymerase" was used. In lines 96, 97,138 and 144, "pol" was used.
Either one should be unified.
5. In line 77, unnecessary space exists.
Author Response
REVIEWER 2
Authors described current status of DNA repair directed precision oncology strategies in epithelial ovarian cancers as a Review. There are many references beyond those listed by the authors, but this is a review and it is up to the authors to decide what to add. Therefore, we believe that the content is mostly acceptable. However, the following points could be revised to make the review more comprehensible to readers.
- To help readers understand, I strongly recommend adding a single, easy-to-understand diagram of the "strategy" using an entire page, as authors described "strategy" in the title.
Author response: As suggested by the reviewer, we have included Figure 2, outlining the current recommended treatment strategy/algorithm for advanced ovarian cancer and the role of PARPi within this.
- There are very many abbreviations. A list of abbreviations should be included for readers.
Author response: A list of abbreviations has now been added at the end of the revised manuscript.
- In Abstract,
"PARP" should be changed to "poly(ADP-ribose) polymerase (PARP)".
Author response: This has been corrected.
- In line 124, "polymerase" was used. In lines 96, 97,138 and 144, "pol" was used.
Either one should be unified.
Author response: Thank you. This has now been corrected.
- In line 77, unnecessary space exists.
Author response: This has now been corrected.